# Enabling Private Investment in Affordable Housing in Nigeria: Lessons from the Experience of the Millard Fuller Foundation Projects in Nasarawa State

Lilian Nwachukwu *, Lucelia Rodrigues and Lorna Kiamba

Department of Architecture and Built Environment, Faculty of Engineering, University of Nottingham, Nottingham NG7 2RD, UK; lucelia.rodrigues@nottingham.ac.uk (L.R.)
* Correspondence: lilian.nwachukwu@nottingham.ac.uk

**Abstract:** Despite the shift to private sector-driven affordable housing in Nigeria for decades, the housing deficit has continued to increase to the disadvantage of low-income families. This paper explores the enabling strategies for stimulating private-driven affordable housing in Nigeria. A case study of the Millard Fuller Foundation projects was undertaken, and semi-structured interviews were administered to 12 residents of the estates and the developer to explore their experience and highlight the considerations for designing appropriate strategies. The data generated were analysed using thematic analysis with the support of Nvivo. This study identifies four major components of construction costs—land, design, materials, and finance—that policy improvement can target to stimulate private investment. It shows that developers are likely to adopt practices that will reduce these costs with repercussions for end-users. Mindful of this, and the concern to make returns on investment, strategies should aim to harmonise both developers' interest and that of the end-users through widespread infrastructural development to make land available in all locations, and an incremental owner-building approach so that end-users can take decisions for their housing. Furthermore, access to National Housing Fund (NHF) mortgages should be enhanced by recognising supplementary incomes in the loan origination procedures.

**Keywords:** affordable housing; private real estate investment; housing market; National Housing Fund; Nigeria

## 1. Introduction

Over the last four decades of shifting to private-driven affordable housing, the housing deficit in Nigeria has been multiplying and is estimated at between 17 and 22 million units (World-Bank 2018, p. 3; Ajayi 2019, p. 232). This figure, when compared to other African countries (Table 1) implies an urgent need to design better strategies for better housing delivery in Nigeria to stem the rising trend in housing deficit. Although these figures are just estimates, the reality of the housing situation in Nigeria is dire, especially in the cities where urbanisation and population growth (Adegun and Taiwo 2011, p. 457; Ajayi 2019, p. 232) have worsened the situation, with the low-income families being the most affected (World-Bank 2018, p. 3; Ajayi 2019, p. 223). Most importantly, the majority of the developments have failed to accommodate the nature of the housing situation; the cost of buying and renting a home has spiralled (Adegoke and Agbola 2020, p. 178), with tenants in rental flats spending up to 60% of their average disposable income on housing (Adedeji et al. 2023, p. 435). Consequently, vacant properties exist alongside homelessness, crowded living, slum, and squatter development (Aliyu and Amadu 2017, p. 150; Moore 2019, p. 205; Adegoke and Agbola 2020, p. 178) despite previous and ongoing efforts to create affordable housing.

**Table 1.** Housing shortfalls in African countries.

| Country | Estimated Population (2016–2019) | Estimated Housing Deficit |
|---|---|---|
| Nigeria | 185–200 million | 8–22 million units |
| Ghana | 28–30 million | 1.7–2.6 million units |
| Kenya | 45–52 million | 2 million units |
| Uganda | 37–43 million | 1.7–2 million units |
| South Africa | 56–58 million | 2.5 million units |
| Ethiopia | 98–103 million | 1.2 million units |

Source: Ajayi (2019, p. 205).

Poverty and unemployment rates are significantly high, the percentage living below the poverty line is now at 40.1% (NBS 2020, p. 5), and the unemployment and underemployment rates have continued to increase. Between now and the fourth quarter of 2020, 33.3% of the Nigerian labour force was unemployed and a further 22.8% was underemployed (NBS 2022). The prevailing low income arising from weak formal job creation, underemployment, low wages, and insufficient skill development (Raschke 2016, p. 7) affect the capacity to fulfil their basic needs, including housing. Furthermore, access to mortgages is constrained, with the mortgage to GDP ratio in Nigeria being only 0.5% as opposed to 31% in South Africa, 2% in Botswana, 2% in Ghana, 77% in the US, 50% in Hong Kong, 52% in Malaysia, and an average of 50% in Europe (Ajayi 2019, p. 224). Access to affordable homes is extremely low when compared to other African countries. Only 25% of the population in Nigeria can access affordable homes, as opposed to Indonesia (84%), Kenya (73%), and South Africa (56%) (Ajayi 2019, p. 223).

In the past, housing in Nigeria was marked by a series of failed public housing programmes, which did not address the housing needs of the low-income population (Olayiwola et al. 2005, p. 2; Ajayi 2019, p. 224; Moore 2019, p. 206). Expectedly, this deficiency has always been supplanted by private efforts such that housing in Nigeria can be deemed as predominantly private-driven, accounting for about 90% of housing in Nigeria (Makinde 2014, p. 51). However, despite being predominantly private-driven, housing has remained elusive to low-income earners. Therefore, in recognition of the vast contributions made by the private sector to housing, the inefficiency of past public programmes, the various factors that presently challenge the government's commitment to housing development (Elegbede et al. 2015, p. 11; Ajayi 2019, p. 234), and the need to harness private resources more effectively, housing policies since 1991 have entrenched an enabled private-driven approach to facilitate housing provision for the low-income segment of society.

The most recent National Housing Policy (NHP) of 2012 planned to tackle the housing deficit through the enabled private sector-driven approach (FGN 2012, p. 67). Although this policy direction, which has been held for four decades, aimed to address the affordable housing problem in Nigeria, the problem persists. Hence, there is both low private investment in and low access to affordable housing (Makinde 2014, p. 62; World-Bank 2018, p. 3; Ajayi 2019, p. 230; Moore 2019, p. 213). Conteh et al. (2020, pp. 1–3) linked low investment in affordable housing to its unprofitable nature, low returns, high risk, and high illiquidity. In Nigeria, these factors manifest in the high transaction cost of land allocation, registration of titles, high interest rates, high cost of materials, and exchange rates (Makinde 2014, p. 62; Ajayi 2019, pp. 234–35). However, a private developer seems to have defied these odds to develop housing that is adjudged the cheapest in Africa for three consecutive years (CAHF 2019, p. 5), challenging the belief that private investment in affordable housing is impossible. Therefore, through an in-depth study of the Millard Fuller Foundation projects, this paper aims to fulfil these objectives:

- To investigate the class of people targeted by the MFF housing projects.
- To determine whether these projects have effectively responded to the housing needs of low-income earners.

- To identify the strategies adopted in constructing and disposing of the MFF affordable housing in Luvu.

Addressing these objectives will highlight key considerations for designing appropriate enabling strategies for private-driven affordable housing in Nigeria.

## 2. Affordable Housing and the Nigerian Context

The absence of a clear definition of affordable housing prevails in many countries; oftentimes, affordable housing is used interchangeably with social housing to imply housing provided with public subsidies. In the UK, the interpretation of this term at different phases in its history is worthy of mention. Generally, UK housing policies over the years were guided by the phrase—"A decent home for every family at a cost within their means", and throughout its history, the elements of this statement have been captured in the policies based on local interpretation and need at the time (Bramley and Karley 2005, pp. 686–87). Hence, in the 1950s, emphasis was on quantity after World War II induced housing shortages; in the 1970s, the focus shifted to quality and eligibility; and recently, the emphasis has been on price vs means (Bramley and Karley 2005, p. 687). Regardless of the interchangeable use of both terms, the Wilson and Barton (2022, p. 7) definition aligns with the above phrase and shows that affordable housing encompasses both social housing and a wide mix of housing intended to satisfy the housing needs of a wide range of low- to middle-income classes.

The use of a wide mix of houses in Wilson and Barton's definition suggests that affordable housing effort offers a range of housing options to cater for different levels of low to middle income and therefore establishes a relationship between housing and people, which is termed affordability (Stone 2006, p. 153). Accordingly, the UN-Habitat (2011, p. 10) defined affordable housing as that, which is adequate in quality and location and does not cost so much that it prohibits its occupants from meeting other basic living costs or threaten their enjoyment of basic human rights. The fact that housing costs should not deny the fulfilment of other basic household needs has become an appropriate measure of affordable housing (termed the residual income approach), and the implication is vividly captured in the Joint Centre for Housing Studies (JCHS 2020, pp. 35–36). According to this survey, 71% of households earning less than USD 15,000 annually in America had a severe cost burden in 2019, leaving them with only USD 225 each month for all non-housing expenses. Over the years, the residual income approach has become more appropriate for identifying cost burdens against the 30% of income that fails to account for the cost of other basic needs or the sacrifices that households are likely to make (Airgood-Obrycki et al. 2022, p. 1).

The residual income approach assesses the residual income of a household, which is the amount left over after paying for housing (Stone 2006, p. 163); it recognises housing as a distinct physical attribute, which when compared with other necessities makes the largest and least flexible claim on the after-tax income of households (Stone 2006). Hence, housing is unaffordable if the residual income cannot meet other non-housing needs like food, healthcare, transportation, childcare, and other necessary expenses at some basic level (Herbert et al. 2018). While the residual income approach is better on account of its ability to define households that are shelter poor (that is, those who cannot fulfil other non-housing needs due to high housing costs), it still presents operational difficulties in determining the basic household expenses, as these vary with household circumstances.

The poor history of housing provision in Nigeria makes the definition of affordable housing somewhat difficult; however, various indicators and the policy direction provide a clue on what affordable housing means in Nigeria. First, the estimated housing deficit is concentrated on low-income families (Ajayi 2019, p. 223; World-Bank 2018, p. 3), who constitute a large proportion of the urban population (Raschke 2016, p. 6; World-Bank 2018, p. 3; CAHF 2020, p. 8). Secondly, there is a significant number of unoccupied houses in the city (Aliyu and Amadu 2017, p. 150; Adegoke and Agbola 2020, p. 178) for sale and rental that urban dwellers cannot afford (Adegoke and Agbola 2020, p. 178), signifying a supply gap in the huge demand for affordable housing. Furthermore, the policy direction

is the provision of social housing for the no income, low-income, and low–middle-income (FGN 2012, p. 66) population, for whom strategies for enhancing access to housing include strengthening the mortgage system and the use of a contributory subsidised National Housing Fund (NHF). Accordingly, the policy categorises these groups (referred to as "the target groups" in this paper) as shown in Table 2.

**Table 2.** Income categorisation and the equivalent in terms of the national minimum wage (NMW).

| Income Category | Earning Capacity | Earning Capacity in Terms of the NMW (NGN 30,000) |
|---|---|---|
| No income | 25% of the NMW | 7500/month |
| Low income | More than 25% of the NMW or equal to NMW | 7600–30,000/month |
| Low–middle income | More than the NMW but less than four times the NMW | 31,000–120,000/month |

Adapted from FGN (2012, p. 66).

The direction of the policy implies a great need for affordable housing because using the NBS (2020, p. 5) and other estimates as a benchmark shows that a conservative estimate of 50%[1] of the population needs affordable housing in the strictest sense. While it may be difficult to measure affordable housing needs in the private sector (formal and informal), in the organised formal sector (namely the public sector) alone, 70 to 80% of civil servants are on grade levels 1 to 10 (Chime 2016, p. 9), earning between NGN 422,566 and NGN 1,535,417 annually (National Salaries, Incomes, and Wages Commission (NSIWC 2019a) and are therefore within the target groups (specifically within low–middle income in Table 2). This range of salary has low affordability for NHF mortgages (see Table 3) and therefore creates a wide affordability gap for the group when compared with the average cost of housing in Nigeria (see Tables 4 and 5). In view of this, and in the absence of a welfare system, one wonders how to enable the private development of houses that can meet the needs of the target groups. Hence, affordable housing in this paper includes housing that is designed to meet the housing needs of the target groups.

**Table 3.** Accessibility challenges to NHF due to low income.

| Grade Level | Monthly Salary | 30% of the Monthly Salary | Loan Repayment over 15 Years | | | Remarks |
|---|---|---|---|---|---|---|
| | | | NGN5 m | NGN8 m | NGN15 m | |
| 3/15 | 38,893 | 11, 668 | 42,192.84 | 67,508.55 | 126,578.52 | Not Qualified |
| 6/15 | 51,708 | 15,512 | Ditto | Ditto | Ditto | Not Qualified |
| 9/15 | 114,309.25 | 34,293 | Ditto | Ditto | Ditto | Not Qualified |
| 12/15 | 145,628 | 43, 688 | Ditto | Ditto | Ditto | Qualified for 5 m |
| 15/15 | 244,498 | 67,349 | Ditto | Ditto | Ditto | Qualified for 5 m |

Adapted from (Udoekanem 2013), NSIWC (2019b).

**Table 4.** Average prices of houses by private developers and the Federal Ministry of Works and Housing.

| Average House Prices from the Private Sector (NGN) | | | | National Housing Programme Houses across Nigeria (NGN) | | |
|---|---|---|---|---|---|---|
| Bedrooms | Abuja | Lagos | Kaduna | Flat in Condominium | | |
| 2 | | 26,670,000 | 16,912,494 | 1 bedroom | 2 bedrooms | 3 bedrooms |
| 3 | 51,740,000 | | 29,637,327 | 7,222,404 | 9,148,378.4 | 13,241,074 |
| 4 | 84,330,000 | 62,740,000 | 35,796,239 | Bungalow | | |
| 5 | 142,680,00 | 87,970,00 | 49,280,015 | 9,268,751 | 12,398,460.2 | 16,491,155.8 |

Adapted from Roland Igbinoba Real Foundation for Housing and Urban Development (RIRFHUD n.d.) and the Federal Ministry of Works and Housing (FMWH 2020).

**Table 5.** Summary of MFF projects, based on data collected from a site visit of the project in 2020.

| ID | Project | Number of Units | Cost (NGN) | Construction Method | Funding | Design |
|---|---|---|---|---|---|---|
| 1 | Fuller estate | 60 | 240,000 | Concrete block and Nigerite-produced drywall | Fuller Centre for Housing (FCH) USA | Studio apartment started in 2007 with the last set completed in 2013 and fully occupied |
| 2 | Camp Luvu I | 13 | 5.9 m | Concrete block construction | Self-funded | Three- and four-bedroom apartments |
| 3 | Aso Fuller estate | 12 | 3 m and 4 m | Concrete block construction | MFF in partnership with Aso savings and loans ltd | One- and two-bedroom semidetached bungalow completed and fully occupied. Started in 2009 and completed in 2010 |
| 4 | Selavip I and II | 36 | 360,000 and 960,000 | Concrete block and Nigerite-produced drywall | Selavip and Etex group | Studio apartments started in 2014 and were completed in 2015 |
| 5 | Grand Luvu I | 268 | 1.65 m, 2.9 m, and 3.9 m | Concrete block construction based on an incremental model [a] | MFF with funding from Reall, UK (loan at 5%) and bought over by FHF | Studio expandable to one-bedroom and one-bedroom expandable to two-bedroom semidetached bungalow started 2015 and completed in 2016 |
| 6 | Camp Luvu II | 32 | 3.6 m and 5 m | Concrete block construction based on an incremental model | MFF with funding from partner Reall | Studio and two-bedroom started in 2020 and ongoing |
| 7 | Grand Luvu II | 400 | 2.9 m and 3.9 m | Concrete block construction | An initiative of FHF completely funded and handed over to it | Studio and two-bedroom semidetached bungalow, which took off in November 2017 and was completed in August 2018 |

Source: Obtained from MFF during a site visit in March 2020. [a] Innovative incremental model is designed for families whose financial status is unable to afford them their desired house; they start with what they can afford, and when their finances improve, they can expand the house to fit their needs. Usually, a studio is designed to be expanded to a one-bedroom apartment and a one-bedroom apartment is designed to be expanded to a two-bedroom apartment.

*The Housing Market and Implications of Private Sector Driven Affordable Housing in Nigeria*

The Nigeria's housing market is influenced by some factors that affect the demand and supply of housing. The population of Nigeria is estimated at 212 million as of 2021 and more than half of this population lives in cities, implying a huge need for housing (CAHF 2021, p. 193); furthermore, about 80% of the urban population lives in substandard conditions (World-Bank 2018, p. 3; Raschke 2016, p. 6), while 58.8% of the urban population lives in slums (CAHF 2020, p. 8), which signifies poor access to housing and huge demand for affordable housing. Generally, poor access to housing is hinged on low disposable income, poor salary, and high cost of living. According to the National Bureau of Statistics, over 40% of the population lives below the poverty line of NGN 137,430/annum (USD 334) (NBS 2021, p. 5). In some countries, like the UK and USA, such people are generally assisted with housing benefits and housing vouchers, respectively, to enable them to pay their rent (Wilson and Barton 2019, p. 31; Perkins 2022). In Nigeria, however, such system is absent and instead, the only form of assistance is a contributory subsidised mortgage system—the National Housing Fund (NHF)—that seems to favour public sector workers and some organised private sector agencies because monthly deductions from the salary source can be made by employers to facilitate contributions of their employees to this fund. Unlike the Housing Provident Fund (HPF) in China that seems to have inspired the NHF, there is no provision of employee housing subsidies to workers to further enhance demand (Yeung and Howes 2006, p. 351). Additionally, the NHF is riddled with access challenges that affect demand and can affect private investment.

Both government and private efforts have failed to address the housing needs of the low-income population (Adegun and Taiwo 2011, p. 458; Ibem 2011, p. 202; Makinde 2014, p. 51), and current housing efforts are still lagging in that respect. The average price of houses from both sectors (Table 4) shows that housing is unaffordable for a majority, considering the prevailing low income. The backlog of supply arising from such practice over the years creates a huge gap on the affordable end of the market, which offers huge investment opportunities if appropriately harnessed, and yet, private investors and even the government are unwilling to accept that serving the low-income market can be profitable (Raschke 2016, p. 8). This is due to some operational challenges present in the market.

Access to land remains a central issue in housing provision since land is governed by land tenure systems (Lawal and Adekunle 2018, p. 2). This system operates within a regulatory framework, which is the Land Use Act of 1978 (LUA). This Act was originally intended to make land available and accessible for developmental purposes (Ghebru and Okumo 2016, p. 6) and vests all land to each state governor, whose consent formalises land transaction and registration. Generally, this process is lengthy and costly (EFInA and FinmarkTrust 2010, p. 37) and impacts negatively on construction-related inputs like materials and finance.

In terms of affordability, land price is volatile and varies greatly with its location and the availability of infrastructure (CAHF 2021, p. 195); the infrastructure stock in Nigeria is 30%, which is below the World Bank benchmark of 70%. It is reflected in insufficient road network linking between commercial centres across the country (International Trade Administration (ITA 2021)). Therefore, the cost of unserviced land without the adequate title in suburban areas ranges from NGN 926/m$^2$ (USD 2.3/m$^2$) to NGN 7716/m$^2$ (USD 19.17/m$^2$). On the other hand, the price of land with primary infrastructure and an adequate title in urban areas ranges between NGN 30,000/m$^2$ (USD 73/m$^2$) and NGN 200,000/m$^2$ (USD 487/m$^2$) (CAHF 2021, p. 194). The prohibitive price of land in cities is generally caused by its higher development value and is further exacerbated by its scarcity caused by speculative sales and the government taking it over for luxurious development (Raschke 2016, p. 6).

The financial intensiveness of housing development is beyond what private savings and retained business earnings can support (Omirin and Nubi 2007, p. 52); developers in the formal sector generally rely on loans from the deposit money banks for housing development despite the challenges of doing so. Access to housing finance from these sources is constrained by higher interest rates (more than 25%) and short tenure (average 3 years) (EFInA and FinmarkTrust 2010, p. 33; CBN 2020, p. 7). Furthermore, it is constrained by the impediments of the LUA because loans are usually secured with a valid Certificate of Occupancy (C of O). The time taken for registering land and the security of title (EFInA and FinmarkTrust 2010, p. 37) arising from poor administrative protocols and poor land records (Omirin and Nubi 2007, p. 52; Adenikinju 2019, p. 26) is the bane of housing development in Nigeria.

Housing development cost is excessively high since more than half of the cost is attributed to materials (Iwuagwu Ben and Iwuagwu Ben 2015, p. 45). Two major factors are responsible for the high materials cost in Nigeria: the construction technology is predominantly cement- and steel-based technology, and these materials are not sufficiently available locally, hence the need to import them. The cost associated with the importation of materials is as high as 50% to 55% (CAHF 2021, p. 195); other related factors like bad roads, the high cost of petrol, import duty, and fluctuating exchange rates (Oke and Emmanuel 2012, p. 104; Ihuah 2015, p. 221; Adenikinju 2019, p. 26; Ajayi 2019, p. 232) drive up housing cost. Furthermore, the government has failed to support local research as expressed in the policy (Uwaegbulam et al. 2019) and local production has not sufficiently scaled to an appreciable level that can compete with imported materials; the industry is monopolised by a few investors due to the high initial capital outlay in setting up factories, stricter licensing rules, high cost of finance, the high number of middlemen, and

inefficient infrastructural facilities (Mojekwu et al. 2013, p. 364; Uwaegbulam et al. 2019; Eboh 2021).

Involving the private sector to realise certain developmental objectives (KMPG 2021, p. 42) is no longer new, and it has become even more popular in the housing sector (Berry et al. 2006, p. 307). Although the footprints of private activities have been visible for a long time in Nigeria, the continuous decline in public resources and government performance has made their engagement in housing formally recognised. Their operation within the existing market is not impressive; however, maximising private resources in affordable housing requires certain considerations that can help investors navigate the challenging environment.

## 3. Materials and Methods

### 3.1. Case Study Area

Nasarawa state is located in Nigeria's North Central region with an estimated 2.6 million people (NASIDA n.d.). It has 13 local government areas of which Karu is the closest, about 5km from Abuja, the Federal Capital Territory (FCT). The relocation of the FCT to Abuja in 1991 and the proximity to the FCT brought sudden economic development to Karu (NSG 2019, p. 17; Oluwadare et al. 2023, p. 2), transforming it from a remote rural settlement to a vibrant urban area (Isma'il et al. 2015, p. 47). As a development corridor to the FCT, it has become the fastest urban area in central Nigeria (Oluwadare et al. 2023, p. 2), with an annual growth rate of 40% due to the influx of migrants from other parts of the country (Oluwadare et al. 2023) and the FCT. In general, the strategic location of the state and its business-friendly climate attracts entrepreneurs and many multinational offices to the state. The state ranks the 11th most accessible state to start a business in Nigeria (NASIDA n.d.; NSG 2019, p. 23). These attributes have made Karu attractive to investors and workers alike and was one of the factors that attracted the development of the MFF estates in the area. Furthermore, compared to Abuja, the relatively low cost of housing is a major attraction to workers in the FCT (Oluwadare et al. 2023, p. 3), and despite these positive attributes, Karu is unplanned and lacking in basic amenities and infrastructure (NSG 2019, p. 18; Oluwadare et al. 2023, pp. 2–3). Housing is mostly rented and few live in their own homes (Isma'il et al. 2015, p. 49). Generally, the most common types of rental residential housing include one-bedroom to three-bedroom flats and single to two or more rooms in a compound. The average rental price for one bedroom flat hovers between NGN 300,000 and NGN 900,000 per annum depending on the location, age, and services provided.

### 3.2. Study Design and Instrument

Case studies are generally used to provide a practical example and context-based knowledge of a situation (Flyvbjerg 2006, pp. 221–22); however, they are criticised for failing to generate generalisable findings based on their limitation of representativeness, which is even worse for a single case (Mariotto et al. 2014, p. 360). Dismissing this criticism as a basis for rejecting the case study approach to research, Siggelkow (2007, p. 20) argued that a single case can be a powerful example capable of provoking powerful insight that might generally fail when considering a general feature shared among many cases. In support of this, Mariotto et al. (2014, p. 364) declared that it is rather more desirable to base the choice of a case on its unusual characteristics, which can generate insights, than on its representativeness. The MFF project possesses unusual features that can generate insight into the current subject; such features as its unique success in delivering affordable housing in Nigeria for over a decade without government support is an intriguing story that can generate insights for advancing the private-driven affordable housing in Nigeria.

This study focused on the experience of the MFF and some residents to draw insights into possible considerations when designing appropriate enabling strategies for effective private sector-driven affordable housing. Hence, both the developer and some resident households of the estates were interviewed. The residents interviewed were mainly federal

government workers who constitute more than 50% of the residents of the MFF estates (this is based on MFF estimates). The choice of participants was influenced by these factors:

- The government housing efforts and strategies seem to focus on and favour government workers; hence, on account of the structured form of their employment, there is a preponderance that relevant information like income, affordability, and access to NHF mortgages can be easily obtained from federal government workers.
- Government workers in Abuja have been driven into the suburbs in search of affordable homes and the MFF houses have provided an affordable destination for them.

In recruiting participants for this study, purposive sampling was adopted. First, the president of the estate was interviewed and asked to recommend participants. Subsequent interviews were conducted with the president of the estate steering members and other interviewees recommending participants, yielding a total of 12 resident households that were interviewed. Two different semi-structured interviews were administered to the participants. The interview for the developer was to ascertain the strategies employed to realise affordable housing and comprised seven questions aimed at understanding the motivation, objective, planning, organisation of resources, implementation, disposition of the houses, and the challenges encountered in the housing projects. On the other hand, the interview for the residents aimed to understand their experience of owning a house and comprised six questions to understand the process of acquiring and paying for the house, the benefits derived, and the challenges experienced in doing so. Accounts from both provide complementary information that can be used for designing appropriate responses to the affordable housing challenges in Nigeria.

The participants were interviewed individually during the researchers' visits to the site of the project in March 2020 and online when the COVID-19 lockdown was enforced in Nigeria. All interviews were audio recorded with the participants' consent. Accordingly, full transcripts of the recorded discussions were produced for each participant in line with the structure of the semi-structured discussion guide; they were analysed using thematic analysis with the support of Nvivo software. Thematic analysis allows for generating themes from the data against identifying them through preconceived themes. In this study, participants' responses were first coded under the question they were responding to; thereafter, concepts conveyed through their narratives were extracted and grouped into subcodes. All data extracts demonstrating the same codes were grouped together, and repeated patterns of meaning in these codes helped to identify the themes. Thereafter, these themes were presented and discussed as complementary information to the responses provided by the MFF.

### 3.3. The History of Millard Fuller Foundation Housing

The Millard Fuller Foundation started as a non-profit house builder in 2006 to tackle the Nigerian housing deficit (Raschke 2016, p. 11). In 2007, its first studio apartments were built with donor funds and sold at zero profit, zero interest rate (see Table 5). Subsequently, the desire to scale up production led Fuller to shift its strategy; currently, it employs short-term project financing (12–24 months) to provide for-profit residential housing developments for the middle- to low-income groups (Raschke 2016, p. 11). The housing projects built by the organisation are in Luvu-Madaki, Masaka, in Nasarawa State of Nigeria, which is only a few Kilometres away from the capital city (Abuja) of Nigeria (see Figure 1) and most workers who are unable to pay for the expensive accommodation in Abuja live and commute to work from there.

The MFF housing projects mainly comprise studio apartments and one-bedroom apartments built as a semidetached bungalow; however, there are other configurations, as shown in Table 5. Some of these houses were delivered in an incremental building fashion similar in concept to the Chilean firm ELEMENTAL (Ferreira n.d.); thus, the foundation offers shell and core houses that are completed on the outside but need additional finishing on the inside. Families or apartment owners can upgrade their studios to one-bedroom

apartments and their one-bedroom apartments to two-bedroom homes. Buyers were assisted in owning a home through a convenient payment plan. As of 2020, when the data were collected, the MFF had finished more than 600 affordable housing units (Table 5) and is scaling up its operation to deliver 600 units.

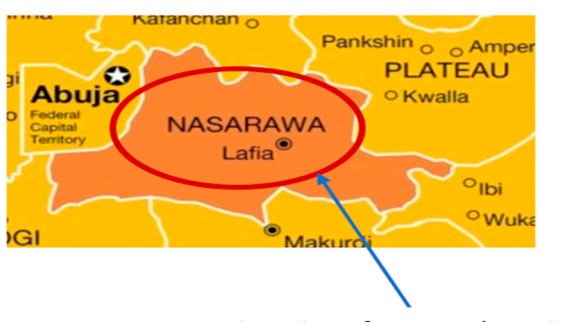

Location of case study projects

**Figure 1.** The proximity of the location of the case projects to Abuja.

## 4. Results

This section presents the results according to the stated objectives. Hence, each theme explored the target end-users, the impact of the housing on the end-users' needs, and the strategies adopted by the MFF to realise the projects. Table 6 shows the profile of the end users who were interviewed and thus provides a description of those targeted by the MFF projects. Apart from the fact that the table provides a clear picture of the housing stress of the residents of the estates (in terms of their income monthly expenditure on housing), five major themes describe how the residents feel that the MFF housing has responded to their housing needs (see Table 7). Finally, six themes describe the strategies adopted to achieve affordable housing. These themes are grouped under predevelopment and development, and the postdevelopment phases as shown in Table 8.

**Table 6.** The profile of interviewed residents.

| Resident | Employment | Monthly Income (NGN) | Type of Accommodation | Number of Persons in the Household | Payment Cost and Plan | Residual Income for Other Household Expenses (NGN) | Cost of House (NGN) |
|---|---|---|---|---|---|---|---|
| 1 | Retired driver of a federal ministry | Not disclosed | Two-bedroom | 2 | Paid in full with proceeds from the sale of land inheritance | Not applicable | 3 m |
| 2 | Federal ministry employee | Not disclosed | Expandable studio and one-bedroom | 5 | Paid 10% of the price and pays NGN 50,000 monthly for 7 years | Not disclosed | 4.55 m |
| 3 | A laid-off staff of a private bank and currently has no job due to age | Not disposed | Two-bedroom | 2 | Borrowed from a friend to pay the initial 10% deposit and makes monthly payments to complete within 5 years | Not disclosed | 3 m |
| 4 | Federal civil servant | 136,000 | Two-bedroom | 3 | Originally ten years but shortened to five years and pays NGN 60,000/month | 76,000 | 3.5 m |
| 5 | Federal civil servant | 125,000 | Two-bedroom | 3 | 50,000/month for 5 years | 75,000 | 3.5 m |
| 6 | Federal civil servant | 40,000 | One-bedroom | 1 | 19,500/month for 15 years | 20,500 | 2.9 m |
| 7 | Federal civil servant | 135,000 | Two-bedroom | 4 | 46,000/month for five years | 89,000 | 4 m |
| 8 | Federal civil servant | 110,000 | Two-bedroom | 4 | 64,000/month for five years | 46,000 | 3.9 m |

**Table 6.** *Cont.*

| Resident | Employment | Monthly Income (NGN) | Type of Accommodation | Number of Persons in the Household | Payment Cost and Plan | Residual Income for Other Household Expenses (NGN) | Cost of House (NGN) |
|---|---|---|---|---|---|---|---|
| 9 | Federal civil servant | 116,000 | Two-bedroom | 5 | Received as a gift | Not applicable | Not applicable |
| 10 | Federal civil servant | 110,000 | Two + one-bedroom | 6 | 90,000/month for five years | 20,000, supplemented by wife (average of 10,000/month) | 6.8 m |
| 11 | Federal civil servant | 100,000 | Two-bedroom | 6 | 50,000/month for five years | 50,000 | 4.2 m |
| 12 | Federal civil servant | 180,000 | 3 | 5 | 80,000/month for five years | 100,000 | 7 m |

**Table 7.** Impact of MFF projects on residents.

| Resident | Acquired from | Ease of Acquisition | Satisfaction Derived | Discomforts or Dislikes |
|---|---|---|---|---|
| 1 | MFF | | Sense of ownership, security (have something to fall back on after retirement), comfort and freedom form harassment of the landlord, safe environment | Poor salary, limited access to mortgage |
| 2 | MFF | | Sense of ownership, no harassment, no paying of rent, sense of relief and comfort | Unfinished building, bad road |
| 3 | MFF | | Ability to own a house with small funds, friendly mode of payment, space around the house for garden, communal outdoor space for recreation | Unfinished building, bad road, short payment period |
| 4 | FHF through workplace cooperative | Bureaucratic processes and the payments involved | No longer having to pay rent in lump sum, sense of ownership, no harassment, sense of pride, ease of payment | Poor quality, not up to standard, small room spaces, low height, unable to fix ceiling fan with such height, wall absorbs water and destroys the painting, a lot of spending on transportation, low wage, more amenities at the city, failure to secure a mortgage, higher monthly payment due to short repayment period, inaccessible road, inadequate water supply |
| 5 | FHF through workplace cooperative | Scheme plan poorly communicated by the government, uncertainty about their ownership status, difficult processing | Better to pay flexible to own than rent, ease of ownership | Rooms are small, will like to make some changes with the open space, spends more time commuting to work, afraid to lose job as a result, problem of water and electricity, bad road |
| 6 | FHF through workplace cooperative | Federal government acquired at cheaper price from the developer and are selling at high prices | Sense of comfort despite the inconvenience of lack of amenities, ease of ownership, security | The spaces are small, spend more on transportation, inadequate water, and no electricity, but that is a national problem |
| 7 | FHF through workplace cooperative | | Ease of ownership and affordability of ownership | Room spaces are small, delineation of spaces is not functional, location of the estate is far, traffic is usually heavy and ends up late at work, short amortisation period and poor communication resulted to higher monthly payment, no water, which adds to the cost of running the house |
| 8 | FHF through workplace cooperative | The payment plan and status poorly communicated; the house was formerly cheap until the federal government bought them | Sense of ownership and the flexible way to own a house, the cost of building from the scratch is high and cannot be achieved with low income, communal space for sports, security of the source of accommodation | Thinks that low-income housing means not adequate provision, location is inaccessible, no adequate water, no electricity |

**Table 7.** *Cont.*

| Resident | Acquired from | Ease of Acquisition | Satisfaction Derived | Discomforts or Dislikes |
|---|---|---|---|---|
| 9 | FHF through workplace cooperative | | | Stress of transportation because of heavy traffic, spends a lot on transportation, lesser productivity, no electricity, stressful and costly to get to place of work |
| 10 | FHF through workplace cooperative | | Better to have mine than pay rent | Longer time on the traffic to get to town |
| 11 | FHF through workplace cooperative | Acquiring through government required lots of documentation and payments | Flexibility to own a house, sense of ownership, cheaper to own than build from the scratch, safe environment | Cheap material for construction, feels unsafe when it rains due to rain penetration, spends time commuting to work, no water, bad road, stress travelling on the road to work due to heavy traffic. Inadequate provision of water, power supply in Nigeria is problematic |
| 12 | FHF through workplace cooperative | | Security, ownership, no longer paying rent, safe environment | Inconveniencing to shorten the payment period because it increases monthly payment and salaries are poor. No amenities like water, it is inadequate for the number of residents |

**Table 8.** Strategies adopted by MFF.

| Construction cost Reduction Strategies | | | |
|---|---|---|---|
| Targets | Method | Reason | Consequences |
| Land cost | Sited project on the outskirts of Abuja | (i) Land is cheaper in Luvu, (ii) There is already an existing relationship with the community, which made the land transaction easier, (iii) Nasarawa state has a good land registration system | Location lacked services and infrastructure, so they were provided by the organisation and the cost was factored into the cost of development. The end-users complained that there is no access road to the project location being far from local transportation and towns. |
| Design cost | The design was limited to bungalows and comprised mainly studios and one- to two-bedroom apartments, the room spaces are compact. | To keep the cost of construction and the materials required for construction as low as possible. | The cost of units was much lower, but the end-users were dissatisfied with the outcome, so some of them had to make some changes to suit their design taste |
| Material cost | Adopted the conventional construction materials in Nigeria, e.g., concrete block, concrete, cement, etc. | To keep the interactive cost of material, construction technology and labour low | Because labour was available for this type of construction, the project resulted in the engagement of the local community, hence employment doubled. |
| Funding cost | Obtained funding from Fuller Centre for Housing, USA, | To reduce cost | It resulted in fewer and slower production, but cheaper apartments sold at no profit and interest (Table 5) |
| | Used soft loan from Reall UK at a 5% interest rate. | To facilitate production at a reduced cost | Increased production but at an interest rate of 5% to cost of construction; the obligation to the loan was eventually paid for with a bulk purchase of the homes by the Family Homes Fund |
| Disposition strategies | | | |
| Selling cost | Set up a flexible payment plan | To assist end-user to pay gradually | Enabled end-users to pay gradually, which suited their variable income and encouraged payment with multiple sources of income |
| | Completed a portion of the house and built the rest up to the concrete oversite | To reduce the cost of construction as well as the selling cost. To enable the end-users to make decisions for their home according to their need and resources. | Encouraged end-users to acquire their dream home in a less stressful manner. |

### 4.1. The Targets of MFF Projects

This section addresses the first objective, which is to identify the targets of the MFF housing project and to ascertain whether MFF is addressing the housing need of the policy target groups (see Table 2). This will help to ascertain whether the policy goal to address the affordable housing crisis through the private sector is possible in Nigeria. Tables 6 and 7 provide information on the profile of the MFF residents. They show that based on the policy definition of the target group (see Table 2), the residents interviewed are within

the low–middle-income range with only four of them beyond this range. Although this number is not representative of the residents of the estates, we can safely assume that judging from the cost of these houses, only those within low–middle-income or above are catered for in this estate. Secondly, it shows that majority of the participants acquired from the government, which came with its challenges (see Table 7).

### 4.2. To Determine If the Project Meets the Need of the Low-Income Population

This objective attempts to determine whether MFF affordable housing meets the need of the low-income population or the residents of the estates. Hence, the interview questions elicited the following information: ease of acquisition, the satisfaction derived, and the turn-offs of living in the estate. The themes emerging from participants' narrative highlight important considerations when designing strategies for enhancing access to housing for the low income.

The information in Table 7 shows that majority of the participants are happy to own a home because of the security it guarantees and the freedom from the harassment that comes with renting, they value the ease of and the flexibility of owning a home that MFF has offered against doing so from the scratch due to low income. However, the high monthly payment towards their housing cost and short repayment period, poor design, including poor communication of ownership plan, location of the estate, inadequate amenities, and bureaucratic process of acquiring a home are some of the challenges of their journey towards owning a home in the MFF.

### 4.3. The Strategies Adopted by MFF

This section describes the strategies adopted by MFF to deliver the affordable homes in Luvu, where they also highlight the challenges to investment for which investors will need intervention to surmount delivering affordable housing. MFF described some of the strategies they adopted to realise affordable housing in Luvu, and their narratives are coded under the relevant interview questions. The interview questions sought to understand actions carried out before and during construction to realise cheaper houses, and those adopted to enable residents to access them. Hence, Table 8 describes the strategies adopted by MFF in two categories—construction cost reduction strategies and disposition strategies.

## 5. Discussion

This study investigated the MFF affordable housing estate at Luvu to draw lessons that may be useful in designing the enabling strategies for private sector-driven affordable housing in Nigeria. To achieve this, three objectives were pursued, namely, to identify the class of people targeted by the project, determine the impact of the project on the residents' housing needs, and identify the strategies adopted by the MFF in delivering the houses. The first objective, which aimed to identify the class of people targeted by the project, served to establish the basis for ascertaining that the measures adopted by the MFF to achieve these houses can be used in designing enabling strategies for private-driven affordable housing in Nigeria. The finding showed that the majority that was interviewed are within the low–middle-income and middle-income group, which means that the strategies adopted in this case may not guarantee affordable housing for the other groups (no income and low-income) mentioned in the policy. Furthermore, the difficulties experienced by this group in terms of residual income after housing cost payment (see Table 6) raise questions about the adequacy of the income classification of the target groups in the present Nigeria circumstances (low minimum wage versus the ever-increasing price of fuel and products in the market, and the exchange rate). It implies that either the minimum wage is increased, or the income capacity ranges of these groups are reclassified to reflect these circumstances. The second objective was intended to ascertain whether the housing needs of the residents were met by the MFF project, using the attributes of housing described in UN-Habitat (2011, p. 10) as a benchmark. Hence, based on the criteria of design, location, basic provisions, and monthly housing payment, the MFF projects have not performed well. With respect to

the strategies employed by the MFF in the development of the houses, these are described in the following themes, reflecting areas during and after construction where developers may require intervention to deliver affordable housing (see Table 8).

*5.1. Land Cost*

Land is crucial for delivering housing (Lawal and Adekunle 2018, p. 3), and its cost and location determine the cost of housing (UN-Habitat 2011, p. 34). Since the land cost can account for a sizeable share of the housing cost, (Woetzel et al. 2014, p. 7; Bah et al. 2018, p. 162), it means that a reduction in the cost of housing can be achieved through the management of the cost components of land. Two essential features of land—affordability and availability—are important for affordable housing and are also interrelated. The availability of land in a good location can affect its affordability and when land is affordable, it is unlikely to become readily available in accessible locations (Bah et al. 2018, p. 109); thus, the cost of providing the infrastructure and services required to enhance accessibility to land are important considerations when siting affordable housing.

The cost of land consists of the cost of acquisition, the cost required to secure land tenure, including the simplicity of the processes and the procedures that are involved (Lawal and Adekunle 2018, p. 3); these considerations naturally move private developers[2] to site housing developments on low-priced land on the urban periphery as a cost-saving strategy for their housing development (UN-Habitat 2011, p. 38).

Therefore, the choice of Luvu as a location for the projects came at a cost for both the MFF and the residents: the MFF bore the cost of providing basic infrastructure and services, which eventually increased the cost of construction that was transferred to the end-users[3] (Bah et al. 2018, p. 109; Makinde 2014, p. 60). Expectedly, this action did not guarantee a pleasant experience for the end-users, who are not only cost burdened but must deal with the inconveniences associated with living far from the towns and commuting to them[4].

Three lessons can be drawn from this theme: First, the lack of affordable land in accessible locations can drive investors to make unhealthy choices for their investments; this highlights the importance of affordable land and the need to improve land value through sites and services programmes (Bah et al. 2018, p. 147). Secondly, despite trading good location for affordability, accessibility has a significant effect on the end-users; this assertion aligns with the analysis of Woetzel et al. (2014, p. 7), which implies that pursuing a reduction in land cost at the expense of good location is detrimental to affordable housing efforts and negatively impacts the end-user's residual income. Finally, the cost of providing services on land has a significant effect on the overall housing cost and the developer will normally transfer the cost to the end-users. Hence, to enable private investment in affordable housing, the provision of infrastructure and services must be pursued by the government.

*5.2. Cost of Design*

Developers who build affordable housing face a lot of hurdles like expensive labour and materials, onerous regulations, and approval, which, together with tight budget constraints, make affordable housing development daunting. The cultural belief that the benefits of good designs should be reserved for those who can afford them is popular, hence it is not uncommon to associate low-income housing with banal and depressing designs (Wright 2014, p. 71). This is demonstrated in the case project where the desire to achieve affordable construction was accomplished through designs that are basic and minimalistic[5].

As much as the MFF achieved a reduction in the cost of construction through minimalistic designs, they may have compromised on quality and comfort in the process as five of the interviewed residents expressed dissatisfaction with the design outcome (see Table 7). Their willingness to make certain changes to their homes[6] suggests that housing is not only about the provision of the basic structure or shell but also about providing the satisfaction



and comfort that the end-users desire for their homes (UN-Habitat 2014, p. 3; Wright 2014, p. 70; Garton et al. 2017, p. 2):

Finally, the actions of the MFF strengthen the already established belief that design variables affect the cost of construction (Seeley 1996, p. 31); so, while efforts may target reducing cost for the developer, they should seek to satisfy the needs of the end-users as well. Achieving harmony between these two interests will lead to a successful affordable housing programme; on the other hand, end-users' tastes vary considerably, so affordable housing strategies may have to lean towards enabling end-user-driven housing initiatives.

### 5.3. Using Conventional Materials vs. Local Counterparts

In Nigeria, up to 55% of the materials cost of construction is due to importation (CAHF 2021, p. 195), and the popular opinion about affordable housing will generally apply. This opinion advocates for the use of local materials and local production to boost large-scale affordable housing development (Acheampong et al. 2014, p. 2; Iwuagwu Ben and Iwuagwu Ben 2015, p. 47) and improve the sustainability of the housing development process (Bah et al. 2018, p. 170). Despite this popular view, the MFF achieved affordability through the use of the conventional materials (cement-based) because they are readily available and accessible in Nigeria (Olajide Olorunnisola 2019, p. 57). In their previous experience[7], they were able to identify important considerations for adopting local-based technologies and materials and they applied the lessons on the Luvu projects.

Therefore, while adopting local or innovative materials and technologies remains a valid approach to affordable housing, and while its full potential is yet to be realised in Nigeria, affordable housing strategies should embody the principles that preserve the attributes of housing development, namely: engaging available labour, boosting employment and local economic prosperity (UN-Habitat 2014, p. 20). Hence, the consideration for adopting local materials in housing should be holistic; research should address the issues of availability of raw materials; the feasibility and viability assessments, which must precede local and large-scale production; and be complemented by corresponding manpower training.

### 5.4. Cost of Funding

Funding is important in housing development because it affects the overall housing cost (Bah et al. 2018, p. 201). The more affordable the funding arrangement, the greater the benefit on affordable housing. This assertion is supported by the MFF experience, which provides valuable lessons on the effect of different funding arrangement on their projects and therefore highlights certain considerations for designing appropriate funding mechanism for affordable housing. There are two major funding arrangements adopted by the MFF. First, free funds or subsidies expressed as donor funding from the Fuller Centre for Housing in the USA helped to realise 60 studio apartments that were sold at no profit or interest[8]; the cost of the studios was, therefore, affordable such that they were quickly sold out and the organisation saw the need to scale-up through a different funding source[9]. On the other hand, the Reall UK loan scaled up the production of housing, but it came with an additional burden of repayment.

Accounting for any economic changes that would have taken place between the period of the development of the first and second studio apartments, the impact of the loan on the selling cost of the houses is visible: at NGN 4.55 m (Table 5), and with the residents not qualified for a mortgage[10], they must find it considerably difficult to make monthly payments and cater for other basic needs (see Table 6).

Two distinct features are important in designing funding arrangements for affordable housing: Firstly, it is indisputable that using free funds led to the delivery of houses that were much more affordable; however, the rate of development (Table 5) compared to when the Reall UK loan was used is low. This means that using free funds or subsidies alone is not a realistic and sustainable funding option because besides funds being limited, the process of awarding them is competitive and they may take time to release in the amounts

that will guarantee speedy delivery of the project (Blumenthal et al. 2016). Table 5 clearly shows that the time taken to deliver 60 studio apartments is much longer than the one-year period required to deliver about 268 units (comprising studio apartments and other types of dwellings) with a loan.

Secondly, although faster production is guaranteed with a loan funding arrangement, the cost of that option is an important consideration when choosing to use it. The interest rate, amortisation period, and the terms for accessing the loan can affect the development cost and subsequently the disposition. Again, the investors are very particular about making quicker returns since it helps discharge them of their loan obligations more quickly. Although this was mitigated by the bulk purchase of its homes by the Family Homes Fund (FHF) (see note 9), which enabled them to pay off the Reall UK loan, planning for affordable housing should be deliberate; this means that improving the end-users' income capacity through appropriate funding mechanism should be considered when planning affordable housing programmes (Blumenthal et al. 2016). Alternatively, bulk buying of houses by the government can help facilitate quicker returns for investors and allow for a flexible disposition of the houses to end-users.

*5.5. Flexible Payment Plan*

It is illogical to invest in affordable housing if the disposition cannot be guaranteed. Investors can only stay in business if they can make returns on their investment, and this can be achieved if end-users have the financial capacity to affect the demand for housing. The low income capacity of the target end-users limits access to housing, which is detrimental to investment. To facilitate access that will guarantee quicker returns for the MMF, a flexible payment plan and an incremental construction approach for the projects were adopted. The original design of a flexible payment plan required the buyers to pay an initial 10% deposit and complete the rest through monthly contribution over a period. Flexible payment pattern was the major attraction for the residents and the fact that they were contributing gradually towards owning a home was enough motivation for them to make such committment[11] (see Table 7). This method of paying for housing aligns with the UN-Habitat (2012, p. 27) strategy for enabling low-income end-users towards homeownership in line with their variable income.

Despite the flexible payment plan, the residents particularly found making the initial deposit difficult due to low income and no savings[12], which clearly denotes low minimum wage; in addition to that, many did not qualify for a mortgage loan[13] and even after five years, most of them had not received a decision on their mortgage application, resulting in further negotiation with the FHF to draw an alternative payment plan. Therefore, most of the participants now pay monthly from their salary over five years despite the inconvenience that it causes them[14]. It is evident in Table 6 that almost all the participants are shelter burdened, paying more than 30% of their salary to housing. Despite this, most of the participants expressed a strong desire to maximise the lifetime opportunity offered by the MFF housing to become a house owner; hence, many are sacrificing other needs for housing, while others are exploring different means like borrowing, using gift donations and supplementary informal incomes to fulfil this obligation.

Three lessons can be drawn from this theme. First, the need to own a house overrides every consideration, as Udechukwu (2008, p. 182) rightly asserted; therefore, despite the inconvenience, participants are willing to make sacrifices to realise this ambition. This means that housing efforts that tilt toward homeownership may be more acceptable than its rental counterpart. Secondly, as much as flexible ownership is a more convenient way of paying for housing for the low-income population, monthly payment plans should be designed affordably in line with their income to ensure that payment to housing cost will not impact negatively on the residual income. Therefore, a longer repayment period may be required to reduce the monthly contribution and thus the cost burden, as resident 7's analogy suggests[15].

Thirdly, from the responses, participants did not qualify for the NHF mortgage loan due to their low income; again, those of them who did not receive any decision to their loan application, given their monthly salary (Table 6), would still not have qualified for a loan (see information in Table 3). In this case, two considerations may apply. First, if the NHF mortgage system is to be considered an enabling mortgage option for the low-income earners, it will need to recognise other supplementary income sources in the loan origination procedures as a strategy for enhancing access to it (Makinde 2014, p. 54). Secondly, the case study proved that flexible payment to housing is possible with direct deduction of payment from the source if appropriately designed and managed to prevent defaulting in payment. Above all, flexible payment plans, whether through mortgage or direct deduction from salary source, should be affordable, which can be achieved by extending payment period to reduce monthly contribution.

*5.6. Incremental Housing Pattern*

The underlying reasons for the adoption of an incremental development approach were to reduce the cost of development for the MFF and the initial cost of acquiring a suitable home for the end-users[16]. The idea is to encourage access for families to their dream homes without the inhibitions of their income since they can gradually build or expand their homes based on their resources and need. In consideration of certain changes made by the residents to their homes (see note 7), incremental housing implemented by the MFF will foreground the needs of the inhabitants rather than the developer since families can take responsibility and care of aspects of housing, which they are in the best position to take in line with the principles of incremental housing (Hasgül 2016, p. 20). In places where this approach was successfully implemented, for example, in Chile, basic, core structures were built, and the individuals built up the space according to their pace and resources (Ferreira n.d.). In the MFF case, this was achieved by building up the core of the house and leaving another portion at the concrete oversite level to allow for expansion as end-users deem fit and at their convenience.

**6. Conclusions**

As the housing deficit continues to increase despite adopting a private sector-driven housing approach, the need for improving the existing strategies for effective private performance becomes imperative. By studying the MFF projects, on account of the desirable attributes, this study drew from the lessons to highlight possible considerations for advancing private sector-driven affordable housing in Nigeria. Hence, it establishes the following:

- That the MFF projects at Luvu catered specifically for low–middle-income and middle-income earners and that unless the National Minimum Wage is improved, strategies adopted in this case may not satisfactorily advance the delivery of affordable housing for this income groups, much less cater for the no income and low-income groups.
- Developers may likely adopt practices that reduce the cost of investment, and these practices generally target the construction cost components like land, materials, and design.
- The decisions made by developers with respect to land and the design of the building may adversely affect end-users' satisfaction; however, an incremental building approach may help to preserve both the developers and the end users' interests in terms of a reduction in the cost of investment and the freedom to make choices for their homes based on their taste and resources, respectively.
- Embarking on a widespread infrastructural development will make development land available in all locations so that affordable housing development can no longer be confined to locations where services are lacking.
- Bulk purchase of housing units can help private developers make quicker returns on their investment and release them from any loan obligation as well as facilitate the disposition of houses on flexible terms to the end users.

- Access to an NHF mortgage can be improved for low-income earners by recognising other supplementary incomes in loan origination procedures.
- Flexible repayment plans should incorporate longer period of payment to reduce monthly payment for housing and increase residual income for end-users.

*Future Research*

Articulating enabling strategies for housing will be beneficial to both the government and private investors. Government resources are constrained; therefore, direct intervention might be impossible; the insights generated in this paper are both revelatory and instructive. Policymakers can assess the existing strategies considering them and can be guided to design specific interventions based on the concerns identified in this paper. This paper also recognises that effective enabling strategies should incorporate efforts for harmonising the needs of private developers and end users; hence, future research should lean towards exploring other harmonising features that will encourage more robust enabling strategies for private-driven affordable housing.

**Author Contributions:** Conceptualisation, L.N.; methodology, L.N., L.R. and L.K.; validation, L.R. and L.K.; formal analysis, L.N.; investigation, L.N.; resources, L.N.; supervision, L.R. and L.K.; writing—original draft preparation, L.N.; writing—review and editing, L.N., L.K. and L.R. All authors have read and agreed to the published version of the manuscript.

**Funding:** This research received no external funding.

**Data Availability Statement:** The data presented in this study are available on request from the corresponding author. The data are not publicly available due to commitment to preserve the confidentiality of participants used for the study.

**Conflicts of Interest:** The authors declare no conflict of interest.

## Notes

1. Fifty percent of the population will account for the no income group who already constitute 40.1% living below the poverty line of NGN 137,430/annum (USD 334) (as per the NBS estimate) and the rest shown in Table 2.

2. "Ok, so let me start with the location, we are essentially located here because land is cheap, there is a historical factor as well; the HFH was working in this community, so we already knew the people, and it was easy to buy land from them and to do other projects but essentially, the bottom line is that land is cheap in this area…". (MFF)

3. "Ok. Because we are working here far away from town, really infrastructure doesn't exist, we have had to provide all the needed infrastructure…. so essentially, we must do everything, and we cost it, and the people at the end of the day, pay for it…, so that cost includes all the infrastructure, land, construction, and a small profit element…". (MFF)

4. "Another challenge is the distance from my workplace. The location of the estate is far, most of the people living in this estate work in Abuja,…so we have to travel a bit and be held in the traffic before we get to the office…" (Resident 7); "However, it is far from the city centre and where I work, and I spend NGN 2600 daily commuting to work every day…". (Resident 4)

5. "…our designs as I said are very basic, eventually, we are working around a single-room model. …and we have typical designs for studio apartments, one-bedroom apartments, we have designs for two-bedroom apartments, we hardly do three-bedroom, it's just because of the cost, we want to stay below the 5-million-naira mark. We have bathroom facilities, we keep it minimal, usually just one bathroom for the house…". (MFF)

6. "The room is not big, it's quite small when you compare what we have here to others particularly in some estates, if I have the opportunity, I will make some changes" (Resident 5); "…the estate developers also take advantage of the fact that their activities are not being supervised to use substandard materials to construct the house. For example, once it rains, my house absorbs water and despite painting the inner walls, the paint peels off. So, we now use wall tiles to the height of the room to make sure that the water doesn't penetrate and affect the furniture in the room…". (Resident 4)

7. "We have in the past used the compressed hard block technology in my last organisation HFH, we did a lot of houses with compressed hard blocks, and we discovered that yes it was cheaper than concrete blocks, but we were paying more for labour. …so at the end of the day, the cost kind of balanced out and we saw that there wasn't that more of an advantage in using compressed hard blocks or stabilised blocks than in using concrete blocks…but essentially we are working with the normal concrete blocks technology that everybody works with, it's known, it's available and also it engages a lot of local labour because we see our work not just as construction but also empowering the community so the more people that can be empowered in the process of the housing delivery, the better for the project…". (MFF)

8    "So, as I mentioned, the first project we did is zero profit, zero interest project with the FCH in the US, which was our first project that was financed entirely by donor funds; then we now got a loan from REALL UK to begin the Grand Luvu project it was a soft loan at 5% interest rate. Grand Luvu II on the other hand, was financed partly with equity, and then we were being paid by FHF in instalments. That's how we were able to finish Grand Luvu II ... So, they sort of took it over, they paid us and just took it. Then the Grand Luvu I which was financed by REALL, they bought it when it was completed". (MFF)

9    "...it's an apartment that cost about 360,000 naira and people wondered: can this be real in Abuja? And they asked: is that the cost of the rent or the cost of the house?...the challenge we had when people came to our doorstep and found it to be so cheap is that the whole of Abuja now ended up on our doorstep and with just donor funding coming in, we could not build more than what we had on hand so most of the people had to be turned back...". (MFF)

10   "ok, initially, we were told that we should pay 10% of the money, after the 10% of the money, they can give you the key to the house and you start paying may be through a mortgage, but for my case, I didn't go through mortgage because, by that time, the mortgage did not accept the percentage I applied for...". (Resident 2)

11   "...but MFF will only ask you for a percentage, when you pay that percentage, you will be given the key to your house without completing your payment and then, they spread out the balance over a period of time and that for me is the greatest help they can give to the less privileged" (Resident 2); "Some work and some don't, so it's not easy for somebody to count 2 million, 3 million easily like that to pay, instead of that, they will have discouragement" (Resident 3); "the salary paid to workers is nothing to write home about but if you are removing every month you may not feel it". (Resident 1)

12   "...Getting the initial payment was the greatest problem because I did not have money saved anywhere...so I cried to a sister and ...she asked me to send my account number, I was like, is this true? ...The initial payment is most people's problem but compared to where you will buy a piece of land and build and enter, it's still better...". (Resident 3)

13   "...I didn't go through the mortgage because they did not accept the percentage I applied for. ...in my own case, we own the whole building...so, we are paying NGN 50,000 in a month because the building was given to us at NGN 4. something million [So you and your husband are contributing to pay?] yes [that means you are also a government worker] my husband is a government worker, but I do business". (Resident 2)

14   "At a point because I took it in 2018 but they're still on the process of the mortgage, so when I now decided to opt out from the beginning of this year and I said OK, thank God, I know I can, at least afford it from my salary and I decided to pay the money for five years and get over it even though it is not going to be easy, but then I said that instead of waiting for mortgage endlessly and afterall, it's a business and they are not giving you free. Though it might be painful, but I just decided to endure it and make the payment within five years. So that is the plan". (Resident 11)

15   "At first, it was going for like 10 years. To make it like easier for us so that the amount they will be deducting from our salary will not be too cumbersome. But in the long run, an issue arose that made them reduce the number of years we're going to pay it for. So, they reduced it to five years...Before, when it was for the period of 10 years, we were paying like NGN 27,000 naira, but now that it has reduced to five years, we are paying like NGN 46,000, something". (Resident 7)

16   "...so essentially, we are in incremental housing, so the idea is that you may not have all the money to do your two-bedroom unit, but you can start off with what you have, start off with the studio apartment, with time, put another room more, with time put another room more ...Actually, one of the designs you might see is the studio apartment and the foundation made already for the additional room, then you also have the one-bedroom unit that's expandable to two-bedroom unit, so essentially that's what we are doing". (MFF)

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
