# Peer review of "Enabling Private Investment in Affordable Housing in Nigeria: Lessons from the Experience of the Millard Fuller Foundation Projects in Nasarawa State"

_jrfm, doi:10.3390/jrfm16090411_

Round 1

Reviewer 1 Report

The paper is well-argued and easy to read. Moreover, there are several interesting takeaways from the paper. One concern that is not easy to address at this point is the potential selection bias due to the way respondents are included (willingness to be interviewed and snowballing). Of particular interest in this respect is whether the vast majority of federal employees are due to this selection (in full or in part) or if this affordable housing scheme was primarily aimed at federal civil servants. A quick fix would be to include a table (from MFF) with summary statistics of households/persons included in the scheme.

Author Response

One concern that is not easy to address at this point is the potential selection bias due to the way respondents are included (willingness to be interviewed and snowballing). Of particular interest in this respect is whether the vast majority of federal employees are due to this selection (in full or in part) or if this affordable housing scheme was primarily aimed at federal civil servants. A quick fix would be to include a table (from MFF) with summary statistics of households/persons included in the scheme- Thank you for flagging this oversight, however, due to the length of this paper, we have provided a short description to explain our reason for the preponderance of government workers in the selection process. This can be found between points 300 to 310

Reviewer 2 Report

See comments intext

The English language quality appears satisfactory in my opinion 

Author Response

(1). Point 39- we have corrected the citation; the error is due to the use of in-text citation function of Endnote.

(2). Point 136- the "it" referred here is the residual income approach and we have clearly defined that. Due to our effort to summarise as much as possible the information in section 2, some improvements have been made, which has affected the original position on the point numbering system (therefore, the oversight you noted, which was originally at point 136 is now at point 131)

(3). Point 256- the author Eboh is an online newspaper reporter, and the article is not paginated, hence, the absence of the page number.

(4). Points 336-337- we have appropriately acknowledged the source of the information and provided a direction to the table for the remainder of the information see points 340 to 342

(5). Point 366- The source of the information in table 5 has been appropriately acknowledged

We have also corrected all cross referencing errors on this paper.

Reviewer 3 Report

Aim of the work is to explore the enabling strategies for stimulating private-driven affordable housing in Nigeria. Some suggestions are provided for improving the research:

first section: check some typos "Error! Reference source not found.."

add the reference for Table 2 and 3 before them in the text

second section: this section should be summarized in the essential concepts and issues regarding the topic in the Nigerian context

third section: the descriptive statistics of the interviewed people are missing

fifth section: it is too long. Maybe a summary should be provided

Author Response

(1). First section: check some typos "Error! Reference source not found.."-The typo errors have all been fixed.

(2). Add the reference for Table 2 and 3 before them in the text- We already referenced table 2 and 3 in text before them. For example, in-text reference to table 2 is on point 148 and table 2 is on point 150, similarly, in-text reference to table 3 is on point 162 and the actual table on point 171.

(3). Second section: this section should be summarized in the essential concepts and issues regarding the topic in the Nigerian context-We have tried to summarise this section without losing information that is relevant to the topic. section 2 is divided into two: the first describes the attributes of affordable housing to provide an understanding of its meaning in Nigeria, and therefore, in the context of the paper. We believe that such description will form the basis for judging the performance of the case study project. The second part of section 2 describes the investment market to establish the basis for understanding the challenges of private investment.

(4). Third section: the descriptive statistics of the interviewed people are missing-point 369 following points to where the participants' profile was provided (i.e., table 6)

(5). Fifth section: it is too long. Maybe a summary should be provided- We recognise that the fifth section is long. However, since this study is qualitative, it was fitting to support the discussions with the appropriate participants' quotes, hence, the length. However, we have tried to summarise the essential details to reduce the word length without loosing the lessons of the study. Thank you.

Round 2

Reviewer 3 Report

The efforts made by the Authors are apprecciated, however some missing relevant literature references is observed. The work can be improved by addying some of those ones: Manganelli, B., Del Giudice, F. P., & Anelli, D. (2022, July). Analysis of the Difference Between Asking Price and Selling Price in the Housing Market. In International Conference on Computational Science and Its Applications (pp. 629-640). Cham: Springer International Publishing + Yeung, S. C. W., & Howes, R. (2006). The role of the housing provident fund in financing affordable housing development in China. Habitat International30(2), 343-356 + Bramley, G., & Karley, N. K. (2005). How much extra affordable housing is needed in England?. Housing Studies20(5), 685-715.

Author Response

he efforts made by the Authors are apprecciated, however some missing relevant literature references is observed. The work can be improved by addying some of those ones: Manganelli, B., Del Giudice, F. P., & Anelli, D. (2022, July). Analysis of the Difference Between Asking Price and Selling Price in the Housing Market. In International Conference on Computational Science and Its Applications (pp. 629-640). Cham: Springer International Publishing + Yeung, S. C. W., & Howes, R. (2006). The role of the housing provident fund in financing affordable housing development in China. Habitat International30(2), 343-356 + Bramley, G., & Karley, N. K. (2005). How much extra affordable housing is needed in England?. Housing Studies20(5), 685-715.

Authors' response: thanks for the recommended references, we have captured some of them in the literature. Points 101 to 107 capture Bramley and Karley (2005) while points 186 to 194 capture that of Yeung and Howes (2006).